# Skinning-free Accurate 3D Garment Deformation via Image Transfer

## Abstract

3D garment animation is key to a wide range of applications including digital humans, virtual try-on, and extended reality. This paper addresses the task of predicting 3D garment deformation from a posed body mesh. Existing learning-based methods mostly rely on linear blend skinning to decompose garment deformation into low-frequency posed garment shape and high-frequency wrinkles. However, due to the lack of explicit skinning supervision, they often produce misaligned garment positions with undesired artifacts during garment re-posing, which corrupt the high-frequency signals. These skinning-based methods consequently fail to recover accurate wrinkle patterns. To tackle this issue, we present a *skinning-free* approach that re-formulates the high-low frequency decomposition by estimating posed (*i*) **vertex position** for low-frequency posed garment shape, and (*ii*) **vertex normal** for high-frequency local wrinkle details. In this way, each frequency modality can be effectively decoupled and directly supervised by the geometry of the deformed garment. Moreover, we propose to encode both vertex attributes as texture images, so that 3D garment deformation can be equivalently achieved via 2D image transfer. This enables us to leverage powerful pretrained image encoders to recover high-fidelity visual details representing fine wrinkles. In addition, we model body-garment interaction via cross-attention between dense body and garment image patches, which refines the naive skinning on sparse joints. Finally, we propose a multimodal fusion to incorporate constraints from both frequency modalities and optimize deformed 3D garments from transferred images. Extensive experiments show that our method significantly improves deformation accuracy on various garment types and recovers finer wrinkles than state-of-the-art methods.

## 1 Introduction

Accurately predicting 3D garment deformation given a posed human body enables a wide range of applications, such as digital humans (Muftić et al., 2005), virtual try-on (Santesteban et al., 2019), and extended reality (Meyer et al., 2001). Traditional works (Provot et al., 1995; Li et al., 2022; Bouaziz et al., 2023) often rely on simulators to generate physically plausible results, however, simulation-based methods are time-consuming and require fine-tuning simulator-specific parameters for each garment, which demands expert knowledge and does not scale to diverse garment types (Luible & Magnenat-Thalmann, 2008; Zhang et al., 2024).

Recently, learning-based methods (Santesteban et al., 2019; Patel et al., 2020; Santesteban et al., 2021; 2022; Pan et al., 2022; Zhao et al., 2023) have received increasing attention thanks to the efficiency and scalability of deep networks. As garment deformation consists of both high-frequency wrinkles and low-frequency posed garment shape, it is challenging for neural networks to jointly optimize both frequency modalities due to the spectral bias (Rahaman et al., 2019; Zhang et al., 2023). To this end, previous works mostly adopt a two-stage decomposition by firstly regressing wrinkles relative to the un-posed garment template, then using linear blend skinning (LBS) to handle the low-frequency garment re-posing. However, due to the lack of explicit supervision on garment skinning, they either assume tight garments and directly skin from the closest body vertex (Santesteban et al., 2019; Patel et al., 2020; Santesteban et al., 2021; 2022), or skin loose garments with virtual joints (Pan et al., 2022; Zhao et al., 2023). Such unsupervised skinning can produce misaligned garment positions and undesired artifacts, which corrupts high-frequency signals. Consequently, existing skinning-based methods often fail to recover fine-grained wrinkles, as illustrated in Figure 1.

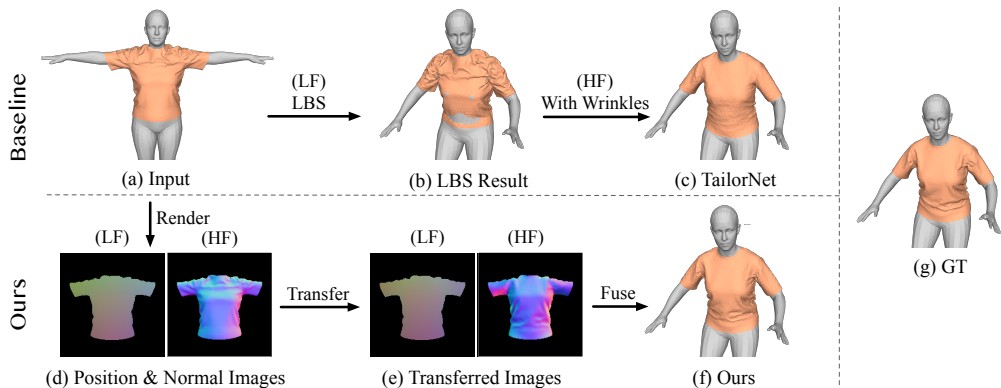

Figure 1: **Illustration of our method.** Given input garment and body meshes (a), previous work (Patel et al., 2020) relies on LBS to generate low-frequency (LF) posed garment shape. However, inaccurate skinning in LBS can produce artifacts and misaligned garment position (b), which corrupts high-frequency (HF) signals and hinders the wrinkle regression (c). In contrast, we decompose frequency modalities using two geometric attributes: vertex positions and normals, which are rendered as 2D texture images (d) and then transferred on pixel intensities (e) to represent garment deformation. After fusing from both modalities, we generate deformed garment with more accurate wrinkles (f).

To tackle the above issues, we present a *skinning-free* method that decomposes high-low frequency modalities with two geometric attributes. Specifically, we propose to directly estimate posed garment **vertex positions** instead of relying on garment skinning. As networks tend to prioritize learning low-frequency signals (Rahaman et al., 2019), with only the position they often generate over-smoothed garment geometry. To recover missing wrinkle details, we further estimate **vertex normals** that better capture local surface bending arising in wrinkles. In contrast to the previous skinning-based methods, our method effectively decouples frequency components and enables explicit supervision for both modalities, which avoids noisy skinning and produces more accurate wrinkles.

Motivated by the recent development of large pretrained image encoders (Dosovitskiy et al., 2020; Caron et al., 2021; Oquab et al., 2023), we propose to project 3D garment geometry onto 2D image space in order to leverage these powerful models. Specifically, we first convert both vertex attributes into RGB colors and render them as texture images from multiple views, so that 3D garment deformation can be equivalently achieved via 2D image transfer, as illustrated in Figure 1. Moreover, predicting garment deformation in image space enables us to recover high-fidelity visual details, *e.g.* wrinkles represented as edges in the normal images, which facilitates to produce deformation of higher perceptual quality. Furthermore, we model fine-grained body-garment interaction via cross-attention between dense body and garment image patches, which refines the naive skinning process over sparse joints. Finally, we fuse priors from transferred images and optimize the overall 3D deformation that aligns with image references.

Our contributions can be summarized as follows. (*i*) We propose a novel skinning-free pipeline for garment deformation with effective high-low frequency modalities decomposition, which avoids noisy garment skinning and facilitates accurate wrinkle regression. (*ii*) We model 3D garment deformation via 2D image transfer, leveraging pretrained image encoders and cross-attention to recover high-fidelity visual details for wrinkles and model fine-grained body-garment interaction. Extensive experiments show that our method noticeably improves prediction accuracy and perceptual quality over state-of-the-art methods.

## 2   RELATED WORKS

**Physics-based Methods.** To generate physically plausible garment animation, physics-based methods either rely on time-consuming simulators (Provot et al., 1995; Bouaziz et al., 2023; Li et al., 2022; Yu et al., 2023), or optimize through physics-inspired losses (Bertiche et al., 2020b; Santesteban et al., 2022; Grigorev et al., 2023). To ensure realism and accuracy, simulator parameters need to be fine-tuned for each garment instance, which can be laborious. Several works propose to estimate these parameters through differentiable simulation (Larionov et al., 2022; Li et al., 2023a) or neural networks (Yang et al., 2017; Clyde et al., 2017), however, the estimation needs to be performed in a

controlled setting with known external factors, which limits their applications. The challenge in data preprocessing thus restricts such method from scaling to diverse garment types.

**Learning-based Methods.** In contrast, learning-based methods (Patel et al., 2020; Santesteban et al., 2019; Pan et al., 2022; Zhao et al., 2023; Zhang et al., 2022) have been developed to achieve superior efficiency and scalability. Pioneered by (Lewis et al., 2000), most works follow to estimate pose space deformation (PSD), namely they adopt LBS to obtain low-frequency posed garment shape, while predicting high-frequency wrinkles in the canonical garment space. Specifically, (Santesteban et al., 2019) directly regresses local vertex displacements using recurrent neural networks. (Patel et al., 2020) proposes to first use mixture models to construct bases of high-frequency deformations, then combine them with narrowed bandwidth kernels. (Zhang et al., 2022) leverages generative models to encode the feasible high-frequency latent space. Similar to our approach, (Lahner et al., 2018; Zhang et al., 2021) uses normal maps to model fine wrinkles. However, they require manually built UV maps and rely on LBS to generate initial normals, which we show in the ablation study that are sub-optimal. While the above works tackle tight garments and directly access body skinning weights, (Pan et al., 2022; Zhao et al., 2023) further extend to loose garments by predicting virtual garment joints to which garments are skinned. However, the prediction of virtual joints can not be explicitly supervised, which can lead to incorrect joint transformations. In summary, existing learning-based methods mostly suffer from noisy skinning that can not be directly supervised. Consequently, the skinning artifacts need to be jointly refined during wrinkle regression, which prevents them from estimating accurate wrinkles. In contrast, we present a *skinning-free* approach, which effectively avoids noisy skinning and facilitates to generate more accurate wrinkles.

**Image-based 3D Representation.** In view of large-scale image datasets and effective pretrained image models, recent works propose to represent 3D geometry in the image space. Most existing works leverage UV mapping as the image representation, which have been widely applied in human pose estimation (Güler et al., 2018), avatar generation (Ma et al., 2021; Li et al., 2023b), and scan registration (Guo et al., 2023). However, they often rely on manual UV unwrapping to produce semantically meaningful islands, which requires expert knowledge and is laborious, thus does not scale to large-scale collections. Alternatively, (Lin et al., 2022; Li et al., 2023b; Xiu et al., 2022; 2023) propose to render multi-view images to automatically establish vertex-to-pixel correspondence. Specifically, (Xiu et al., 2022; 2023) learn to generate 3D clothed humans by integrating from estimated normal images. (Lin et al., 2022) encodes 3D character animation via ultra dense pose images. (Li et al., 2023b) designs Gaussian maps rendered from template human meshes to encode parameters for Gaussian splatting. Unlike all above works that consider only a single image source, we observe that accurate garment deformation requires effectively fusing *multiple* image domains with *mixed* frequency modalities, which is achieved via a novel pipeline as will be introduced below.

## 3 METHOD

To avoid skinning artifacts and produce more accurate wrinkles, we present a novel skinning-free pipeline, as shown in Figure 2. Given a garment template mesh $\bar{\mathbf{M}}_g = \{\bar{\mathbf{V}}_g, F_g\}$ where $\bar{\mathbf{V}}_g \in \mathbb{R}^{N \times 3}$ denotes vertex positions and $F_g \in \mathbb{Z}_+^{F \times 3}$ denotes triangle faces, we aim to estimate its deformed mesh $\hat{\mathbf{M}}_g = \{\hat{\mathbf{V}}_g, F_g\}$ conditioned on the posed body mesh $\mathbf{M}_b = \{\mathbf{V}_b, F_b\}$. Instead of directly regressing 3D vertex displacements, we propose to model garment deformation in 2D image space. Specifically, we first render RGB images $\bar{\mathcal{P}}_g^s$ and $\bar{\mathcal{N}}_g^s \in \mathbb{R}^{H \times W \times 3}$ that respectively encode vertex positions and normals of the garment template from each view $s \in \{\texttt{front}, \texttt{back}\}$. Similarly for the posed body mesh we render corresponding images $\{\mathcal{P}_b^s, \mathcal{N}_b^s\}$. We then develop an image transfer network to generate transferred images $\hat{\mathcal{P}}_g^s$ and $\hat{\mathcal{N}}_g^s$, which describe posed garment shape and wrinkle details respectively (Section 3.2). Finally, we propose a multimodal fusion process to leverage priors of both modalities and optimize 3D deformed garment mesh from transferred images (Section 3.3).

### 3.1 IMAGE RENDERING

We represent 3D garment and body meshes as 2D images rendered from *multiple views*, where the intensities of pixels encode the garment geometry, *e.g.* positions or normals of vertices. Specifically, to generate such an image for a deformed mesh, we use its vertex positions and normals to color the corresponding vertices on the *template mesh*, then render the results from both front and back

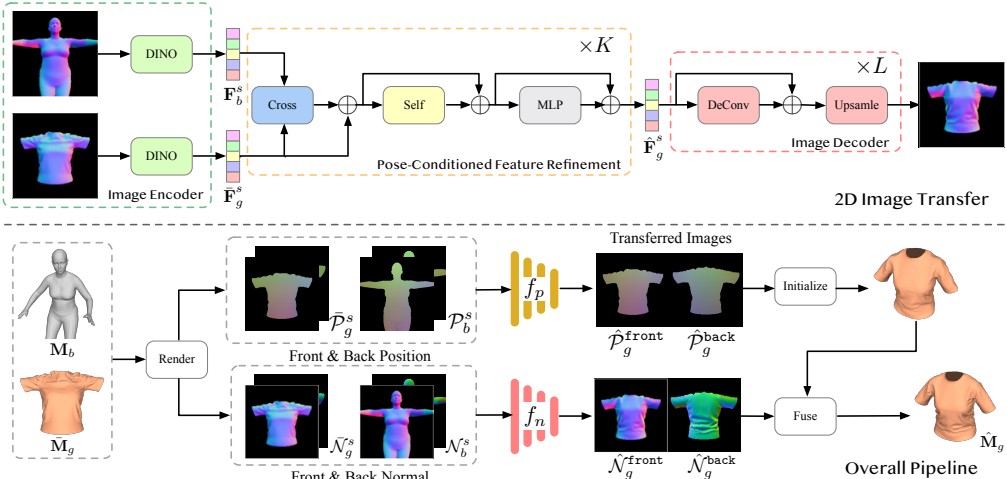

Figure 2: **Overview of our method.** Given the input garment template $\bar{\mathbf{M}}_g$ and posed body mesh $\mathbf{M}_b$, we first render position and normal images for the garment $\{\bar{\mathcal{P}}_g^s, \bar{\mathcal{N}}_g^s\}$ and body $\{\mathcal{P}_b^s, \mathcal{N}_b^s\}$ from each view $s$, aiming to project the 3D garment onto the image space. Next, we transfer position images in $f_p(\cdot)$ and normal images in $f_n(\cdot)$, where the two networks have the same architecture as shown in the top row (taking front normal images as an example). Finally, we initialize the posed garment mesh from transferred position images $\hat{\mathcal{P}}_g^s$ and recover missing wrinkle details by fusing from normal images $\hat{\mathcal{N}}_g^s$ to obtain the deformed garment $\hat{\mathbf{M}}_g$. "$\oplus$" denotes residual connection.

views. In this way, for different deformations of the same mesh, the rendered pixel values will be different while the image silhouette *remains the same*, as we always project the template mesh onto the image space. Taking the garment as an example, given vertex positions $\mathbf{V}_g$ for a deformed mesh, we compute the corresponding vertex normals $\mathbf{N}_g \in \mathbb{R}^{N \times 3}$ and linearly rescale their values to fit RGB colors, *i.e.* within the range $[0, 1]$. We then render the images from each view with a perspective camera of known transformation matrix as:

$$\mathcal{P}_g^s = f_r^s(\text{RGB}(\mathbf{V}_g); \bar{\mathbf{M}}_g), \qquad \mathcal{N}_g^s = f_r^s(\text{RGB}(\mathbf{N}_g); \bar{\mathbf{M}}_g) , \qquad (1)$$

where $\text{RGB}(\cdot)$ represents the linear rescaling function that maps positions or normals to RGB values, and $f_r^s(\cdot; \bar{\mathbf{M}}_g)$ represents the renderer function from view $s$ with the constant template $\bar{\mathbf{M}}_g$. Similarly, we can obtain images for the posed body as $\mathcal{P}_b^s$ and $\mathcal{N}_b^s$, where we use the same cameras as for the garment to capture only relevant body areas. We illustrate in Appendix B for an rendering example.

The above rendering configuration has several advantages compared to alternatives. First, we can automatically establish vertex-to-pixel correspondence through perspective projection, thus do not require manually built UV parameterization (Lahner et al., 2018) or sew patterns (Pietroni et al., 2022; Li et al., 2023a). The rendered images also retain the canonical garment shape, which facilitates the image feature extraction. Second, instead of directly projecting the deformed mesh, we project the template mesh throughout rendering, which avoids introducing new self-occlusions during garment deformation. Moreover, we use front and back views to efficiently capture most visible garment vertices, which also provide sufficient constraints to infer non-visible vertices at side views. After rendering all images, we can estimate 3D garment deformation via 2D image transfer, as will be introduced in the following section.

### 3.2 2D IMAGE TRANSFER

As the garment geometry can be fully represented by the position and normal images, we formulate 3D garment deformation as an image transfer task, *i.e.* we aim to transfer from the initial images $\{\bar{\mathcal{P}}_g^s, \bar{\mathcal{N}}_g^s\}$ representing the garment template to the target images $\{\mathcal{P}_g^s, \mathcal{N}_g^s\}$ representing the deformed garment, conditioned on the posed body images $\{\mathcal{P}_b^s, \mathcal{N}_b^s\}$ as:

$$\hat{\mathcal{P}}_g^s = f_p(\bar{\mathcal{P}}_g^s, \mathcal{P}_b^s), \qquad \hat{\mathcal{N}}_g^s = f_n(\bar{\mathcal{N}}_g^s, \mathcal{N}_b^s) , \qquad (2)$$

where $\hat{\mathcal{P}}_g^s$ and $\hat{\mathcal{N}}_g^s$ are the estimated posed position and normal images, respectively. The position transfer network $f_p(\cdot)$ and normal transfer network $f_n(\cdot)$ have the same architecture (as in Figure 2).

Each network contains three consecutive modules: (*i*) an image feature encoder that extracts visual features of garment and body geometry, (*ii*) a pose-conditioned feature refinement module that injects pose condition and models fine-graiend body-garment interaction, and (*iii*) an image decoder that decodes the transferred images. Below, we will introduce each module in details.

**Image Feature Encoder.** We forward each image input to a pre-trained vision transformer DINO (Caron et al., 2021) to encode patch-wise tokens of image features $\bar{\mathbf{F}}_g^s, \mathbf{F}_b^s \in \mathbb{R}^{M \times D}$ for garment and body respectively, where $M$ represents the number of tokens and $D$ represents the feature dimension. Compared with other encoders like ImageNet-pretrained ResNet (He et al., 2016), DINO can effectively encode detailed structural and visual information through attention on salient image contents, which is beneficial for generating fine wrinkles. We further show its efficacy in Section 4.5.

**Pose-Conditioned Feature Refinement.** We refine image features to introduce pose priors in $K$ transformer blocks. In each block, motivated by (Wang et al., 2023), we model body-garment interaction by first computing the multi-head cross-attention (Vaswani et al., 2017) between the garment feature $\bar{\mathbf{F}}_g^s$ and the body feature $\mathbf{F}_b^s$ to generate the pose-conditioned garment feature. In contrast to the traditional skinning process that computes the skinning weights with respect to sparse joints, we learn to model the *dense* correlation between image patches, which can capture fine-grained body-garment interaction. Furthermore, we follow the vanilla transformer structure (Vaswani et al., 2017) and continue to forward the feature into a self-attention layer followed by a multi-layer perceptron (MLP) to generate the refined garment feature $\hat{\mathbf{F}}_g^s$.

**Image Decoder.** Finally, $\hat{\mathbf{F}}_g^s$ is rearranged spatially to form a 3D tensor corresponding to the 2D image feature map and forwarded to an image decoder to generate transferred images. The image decoder consists of residual blocks of 2D convolution layers, followed by transposed convolution layers to upsample the spatial resolution.

**Training Objectives.** We train each network using the masked L1 loss as:

$$\mathcal{L}_p = \sum_s ||\bar{\mathcal{S}}_g^s \odot \hat{\mathcal{P}}_g^s - \bar{\mathcal{S}}_g^s \odot \mathcal{P}_g^s||_1, \quad \mathcal{L}_n = \sum_s ||\bar{\mathcal{S}}_g^s \odot \hat{\mathcal{N}}_g^s - \bar{\mathcal{S}}_g^s \odot \mathcal{N}_g^s||_1 , \tag{3}$$

where $\mathcal{P}_g^s$ and $\mathcal{N}_g^s$ represent ground truth position and normal images, $\bar{\mathcal{S}}_g^s$ represents the silhouette of the garment template to mask for valid pixels, and $\odot$ represents pixel-wise multiplication. Note that we independently model each modality regardless of their consistency constraints, as we observe that mixing frequency components during image transfer leads to inferior accuracy (as compared in Section 4.5). Alternatively, we opt to fuse position-normal correlation via explicitly optimization.

### 3.3 3D Multimodal Fusion

While we can obtain vertex positions of the deformed garment solely from the position images $\hat{\mathcal{P}}_g^s$ of both views, we observe two major issues of such an approach: (*i*) since the high frequency wrinkle details are often reflected by relatively small position changes, it is hard for the position transfer network $f_p(\cdot)$ to capture such subtleties, thus leading to an over-smoothed mesh, (*ii*) although images from front and back views cover most of the garment, the positions of non-visible vertices at side views can not be directly obtained from these images. In this section, we propose a multimodal fusion process to address both issues. The key idea is to incorporate high-frequency wrinkle details recorded by the normal images to refine the over-smoothed mesh initialized from the position images, while using the edge and surface priors to constrain the non-visible vertices. Specifically, we aim to optimize the optimal deformed vertex positions $\mathbf{V}_g^\star$ that aligns with both image observations as:

$$\mathbf{V}_g^\star = \arg \min_{\hat{\mathbf{V}}_g} \sum_s (||f_r^s(\hat{\mathbf{V}}_g) - \hat{\mathcal{P}}_g^s|| + ||f_r^s(\hat{\mathbf{N}}_g) - \hat{\mathcal{N}}_g^s||) , \tag{4}$$

where $f_r^s(\cdot)$ is the renderer function in Eq.(1) omitting constants, and $\hat{\mathbf{N}}_g$ are vertex normals computed from the vertex positions $\hat{\mathbf{V}}_g$. The optimization consists of two stages, where we first initialize garment mesh from position images and then refine it with normal images to recover fine wrinkle details.

**Vertex Position Initialization.** We initialize vertex positions from position images based on their visibility under the perspective projection. For visible vertices, we simply interpolate their corresponding pixel values in the transferred position images to initalize their positions. For non-visible vertices especially at side views, we initialize them by linearly interpolating from the closest front and

back visible vertex pairs. To correct the linear interpolation and ensure a smooth boundary between two types of vertices, we smooth the results by minimizing the edge length loss $\mathcal{L}_e$ as:

$$\mathcal{L}_e = \frac{1}{|\mathcal{E}|}\sum_{\{i,j\}\in\mathcal{E}}(\|\hat{\mathbf{V}}_g[i] - \hat{\mathbf{V}}_g[j]\| - \|\bar{\mathbf{V}}_g[i] - \bar{\mathbf{V}}_g[j]\|)^2 , \tag{5}$$

where $\mathcal{E}$ represents the index set of all edges defined by the garment faces $F_g$, $\hat{\mathbf{V}}_g$ and $\bar{\mathbf{V}}_g$ represent estimated and template mesh vertices, respectively. Moreover, we impose a regularization loss $\mathcal{L}_{rv}$ to penalize the $L_2$ distance on displacements of visible vertices, in order to align with the position images. The overall loss for this stage can be summarized as $\mathcal{L}_e + \lambda_{rv}\mathcal{L}_{rv}$, with loss weight $\lambda_{rv}$.

**Vertex Normal Fusion.** To amend high-frequency wrinkle details on the position-initialized vertices, we fuse normal predictions onto them by minimizing the normal rendering loss $\mathcal{L}_r$ defined as the second term in Eq.(4), and then smooth the surface normals using a normal consistency loss $\mathcal{L}_{rn}$ as:

$$\mathcal{L}_r = \sum_s||f_r^s(\hat{\mathbf{N}}_g) - \hat{\mathcal{N}}_g^s|| \qquad \mathcal{L}_{rn} = \frac{1}{|\mathcal{E}|}\sum_{\{i,j\}\in\mathcal{E}}(1 - \hat{\mathbf{N}}_g[i]^T\hat{\mathbf{N}}_g[j]) . \tag{6}$$

Similar to the initialization stage, we impose the edge length loss $\mathcal{L}_e$ to penalize irregular rim contours and include the vertex displacement regularization $\mathcal{L}_{rv}$ on all vertices. Finally, to penalize garment-body collision, we impose a collision loss by penalize the penetration distance as:

$$\mathcal{L}_c = \frac{1}{N}\sum_i \max(0, -\text{SDF}(\hat{\mathbf{V}}_g[i], \mathbf{M}_b)) , \tag{7}$$

where $\text{SDF}(\cdot)$ represents the vertex-to-mesh signed distance. The overall optimization objectives for normal fusion can be summarized as:

$$\mathcal{L} = \mathcal{L}_r + \lambda_{rn}\mathcal{L}_{rn} + \lambda_e\mathcal{L}_e + \lambda_{rv}\mathcal{L}_{rv} + \lambda_c\mathcal{L}_c , \tag{8}$$

where $\lambda_{rn}, \lambda_e, \lambda_{rv}, \lambda_c$ are hyper-parameters for weights of losses.

## 4 EXPERIMENTS

### 4.1 DATASETS

We evaluate ours and baseline methods on two benchmarks: VTO (Santesteban et al., 2019) and TailorNet (Patel et al., 2020) datasets.

**VTO.** The VTO dataset (Santesteban et al., 2019) provides two types of garments: tight "t-shirt" and loose "dress". Each garment is draped onto a SMPL (Loper et al., 2015) human body with ground truth deformations simulated in the ARCSim (Narain et al., 2014) simulator. We follow (Pan et al., 2022) to use 4 clips (`01_01`, `111_02`, `55_27` and `91_36`) of medium body shape ($\boldsymbol{\beta} = 0$) and unseen poses for testing and the remaining 49 clips for training.

**TailorNet.** Since the VTO dataset only contains upper garments, we further adopt the TailorNet (Patel et al., 2020) dataset to test on lower garments: tight "pants" and loose "skirt", with ground truth deformations simulated by the Marvelous Designer. We follow (Pan et al., 2022) to use the medium body shape and garment style ($\boldsymbol{\beta} = 0$, $\boldsymbol{\gamma} = 0$) split, and adopt 2 clips (`005`, `010`) of unseen poses for testing and the remaining 16 clips for training.

**Metrics.** Following (Pan et al., 2022; Zhao et al., 2023), we evaluate all methods on three metrics: Root Mean Squared Error (RMSE), Hausdorff distance (Attouch et al., 1991), and spatio-temporal edge difference (STED) (Vasa & Skala, 2010). RMSE and Hausdorff distance assess the prediction accuracy, while STED evaluates the perceptual quality of deformation. Specifically, RMSE calculates the average Euclidean distance between vertices and Hausdorff distance measures the maximum distance between the closest vertex pairs, both in terms of the predicted and ground truth meshes. In addition, we use STED to assess the perceptual similarity of the deformation, which measures the relative edge differences between the predicted and ground truth meshes across each test clip.

### 4.2 IMPLEMENTATION DETAILS

We implement our models in PyTorch (Paszke et al., 2017) and perform all experiments on a single NVIDIA RTX 3090 GPU. We render all images at $256 \times 256$ pixels using differentiable renderer from

Nvdiffrast (Laine et al., 2020) and normalize deformed garment and posed body vertices with the global rotation and translation from the human pose. For the image transfer network, we fine-tune the last two layers of the DINO encoder, along with $K = 4$ transformer blocks for the feature refinement module. We train the model using the Adam (Kingma & Ba, 2014) optimizer for 100K iterations, and set the learning rate to $1 \times 10^{-4}$. For multimodal fusion, we use the same optimizer with a learning rate of $1 \times 10^{-3}$ and optimize for 100 steps in both the initialization and normal fusion stages. For the hyperparameters of losses, we set $\lambda_{rv} = 0.02$, $\lambda_e = \lambda_c = 100$, and $\lambda_{rn} = 0.001$ on t-shirt and 0.01 on other garments based on their scales. We include the detailed model architecture in Appendix C, and report inference time comparison in Appendix D.

## 4.3 QUANTITATIVE EVALUATION

Following (Pan et al., 2022; Zhao et al., 2023), we report all metrics by training and testing on each garment instance to ensure a fair comparison. In Table 1, we present the results on the VTO dataset, where metrics for baselines (Patel et al., 2020; Santesteban et al., 2019; Pan et al., 2022; Zhao et al., 2023) are evaluated by (Pan et al., 2022; Zhao et al., 2023) following the same test configuration. For (Santesteban et al., 2021), we use its official weights and evaluate the results on our test split. We observe that our method achieves the best accuracy against all skinning-based methods (Patel et al., 2020; Santesteban et al., 2019; Pan et al., 2022; Zhao et al., 2023) thanks to the proposed skinning-free pipeline that avoids artifacts of noisy garment skinning. In particular, we outperform (Pan et al., 2022; Zhao et al., 2023) on loose garments, without the need to estimate additional virtual joints or anchors to facilitate garment skinning. Compared to the physics-based method (Santesteban et al., 2021) that refines with self-supervised physics losses, our method generates more accurate wrinkle details that better align with the ground truth data. Moreover, thanks to the capability of perceptual learning in image models (Amir et al., 2021), our method achieves improved perceptual quality (lower STED values) on deformed garments.

Table 1: **Quantitative comparison on the VTO Dataset.** Best results are highlighted in **bold**. Our method achieves superior deformation accuracy and perceptual quality compared to state-of-the-art skinning (Patel et al., 2020; Santesteban et al., 2019; Pan et al., 2022; Zhao et al., 2023) and physics (Santesteban et al., 2021) based methods on both tight and loose garments.

| Methods | Dress | | | T-shirt | | |
|---|---|---|---|---|---|---|
| | RMSE ↓ | Hausdorff ↓ | STED ↓ | RMSE ↓ | Hausdorff ↓ | STED ↓ |
| TailorNet (Patel et al., 2020) | 22.95 | 76.80 | 0.0757 | 9.90 | 27.02 | 0.0418 |
| Santesteban (Santesteban et al., 2019) | 20.96 | 87.01 | 0.0745 | 10.25 | 29.56 | 0.0449 |
| Santesteban (Santesteban et al., 2021) | 21.07 | 87.98 | 0.0620 | 9.97 | 25.64 | 0.0335 |
| VirtualBones (Pan et al., 2022) | 19.91 | 83.39 | 0.0722 | 10.52 | 31.51 | 0.0452 |
| AnchorDEF (Zhao et al., 2023) | 16.05 | 74.20 | 0.0493 | 6.25 | 26.31 | 0.0262 |
| Ours | **13.40** | **61.73** | **0.0407** | **4.66** | **20.89** | **0.0205** |

In Table 2, we report the results on the TailorNet test set. For metrics on pants, we use the baseline results reported by (Pan et al., 2022). Since (Santesteban et al., 2019; Pan et al., 2022) do not release the training code or pretrained weights for the skirt split, we are unable to evaluate their results and thus only compare with (Patel et al., 2020; Grigorev et al., 2023) using their official weights and default physics parameters. We observe that our method consistently outperforms the skinning-based method (Patel et al., 2020) when applied to lower garments, and that the physics-based method (Grigorev et al., 2023) cannot generate accurate results without fine-tuning cloth parameters.

Table 2: **Quantitative comparison on the TailorNet Dataset.** Best results are highlighted in **bold** and and inapplicable results are marked with "-". Our method consistently generates more accurate results than baseline methods on lower garments.

| Methods | Pants | | | Skirt | | |
|---|---|---|---|---|---|---|
| | RMSE ↓ | Hausdorff ↓ | STED ↓ | RMSE ↓ | Hausdorff ↓ | STED ↓ |
| TailorNet (Patel et al., 2020) | 4.84 | 14.46 | 0.0127 | 7.76 | 16.28 | 0.0162 |
| Santesteban (Santesteban et al., 2019) | 4.91 | 14.87 | 0.0129 | - | - | - |
| VirtualBones (Pan et al., 2022) | 4.76 | 18.75 | 0.0166 | - | - | - |
| HOOD (Grigorev et al., 2023) | 5.53 | 17.25 | 0.0175 | 9.18 | 18.85 | 0.0194 |
| Ours | **4.03** | **13.55** | **0.0114** | **5.38** | **14.06** | **0.0150** |

## 4.4 QUALITATIVE EVALUATION

We present qualitative results on the VTO and TailorNet test sets in Figures 3 and 4, respectively. Our method can generate 3D deformed garments with finer wrinkle details and more accurate fold patterns. In comparison, the skinning-based method (Patel et al., 2020) can not recover accurate wrinkles in challenging cases, thus generating over-smoothed geometries. Moreover, physics-based methods (Santesteban et al., 2021; Grigorev et al., 2023) model garment deformation via intrinsic cloth materials and estimate them solely from the garment geometry. Since cloth materials can have a global effect on the deformation, inaccurate materials estimation can lead to misaligned posed shapes or even unrealistic behaviors, *e.g.* over-stretched skirts. Finally, we show in Figure 5 that accurate estimation produces penetration-free results, which well align with the underlying body motions.

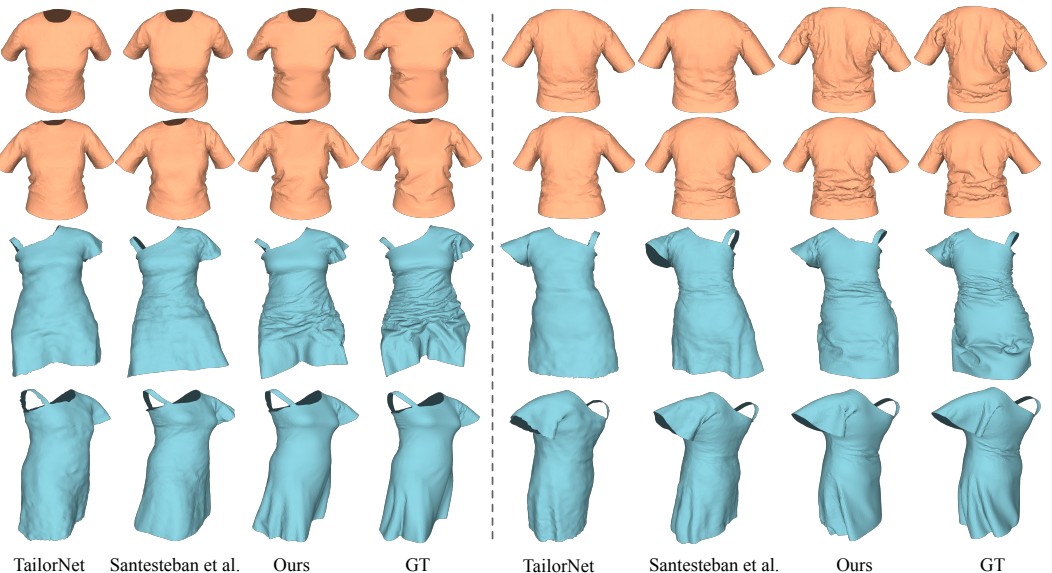

| TailorNet | Santesteban et al. | Ours | GT | TailorNet | Santesteban et al. | Ours | GT |

Figure 3: **Qualitative results on the VTO test set.** Our method produces more accurate wrinkles and folds on loose garments compared to both skinning (Patel et al., 2020) and physics (Santesteban et al., 2021) based baselines.

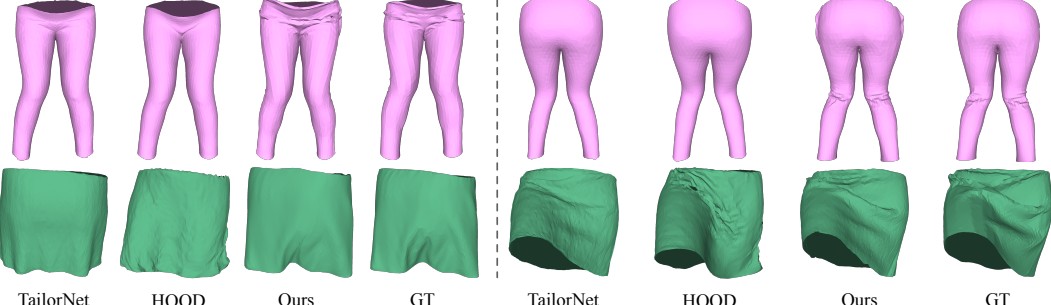

| TailorNet | HOOD | Ours | GT | TailorNet | HOOD | Ours | GT |

Figure 4: **Qualitative results on the TailorNet test set.** Our method consistently produces more accurate deformation on lower garments than skinning (Patel et al., 2020) and physics (Grigorev et al., 2023) based methods.

## 4.5 ABLATION STUDY

**Effects of Skinning-free Approach.** To show the efficacy of our skinning-free approach, we compare in Table 3 with two LBS-based variants (both using skinning weights from nearest body vertices) equipped with our image transfer modules: (*i*) we supervise with GT transferred images in the canonical space, which are generated via inverse LBS (Canonical Image + LBS), and (*ii*) we refine from input images of LBS re-posed garments (LBS + Image Refine), analog to (Lahner et al., 2018; Zhang et al., 2021). We observe that both variants produce inferior results compared to our skinning-free approach, as inaccurate skinning can produce noisy artifacts, which corrupt high-frequency signals in either GT or input images and thus hindering learning correct wrinkle patterns.

Table 3: **Effects of the skin-free approach.** We show that introducing linear blend skinning in the deformation pipeline leads to inferior performance than skinning-free method.

| Methods | Dress | | | T-shirt | | |
|---|---|---|---|---|---|---|
| | RMSE ↓ | Hausdorff ↓ | STED ↓ | RMSE ↓ | Hausdorff ↓ | STED ↓ |
| Canonical Image + LBS | 18.50 | 78.25 | 0.0625 | 6.85 | 25.30 | 0.0295 |
| LBS + Image Refine | 15.75 | 68.40 | 0.0493 | 6.01 | 24.15 | 0.0242 |
| Ours (Skinning-free) | **13.40** | **61.73** | **0.0407** | **4.66** | **20.89** | **0.0205** |

**Effects of Image Transfer Modules.** To verify the effects of key modules in the image transfer network, we compare several alternatives to the current designs: (a) replacing body-garment cross-attention with simply adding the body and garment features (w/o Body Attn.) (b) replacing the DINO encoder with ResNet-50 (ResNet Encoder), and (c) adding cross-attention between two modalities (w/ Corss Modal), as illustrated in Table 4 and Figure 6. We find both (a) and (b) lead to smoothed garment geometry, showing the efficacy of the body-garment cross-attention for modeling fine-grained body-garment interaction, as well as the DINO encoder for extracting detailed garment structural information. Moreover, we empirically observe that mixing frequency signals during image transfer like (c) results in slightly inferior accuracy. To this end, we choose to separately tackle each modality. In addition, we compare with the image representation of automatically generated UV maps via xatlas in Figure 6, and observe that such images contain a large number of islands and therefore destroys garment shape priors, which is not conducive to the pretrained encoders and thus does not benefit wrinkle estimation. More ablation studies for image transfer are in Appendix E.

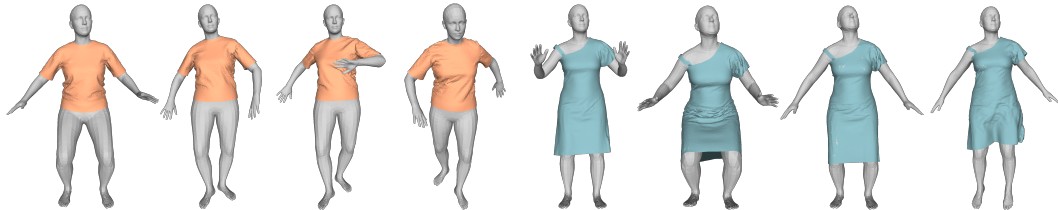

Figure 5: **Qualitative results with human motions.** Our method generates accurate and plausible garment deformations for a sequence of unseen human poses. Moreover, the deformed garments are temporally consistent and collision-free. We show more results in the supplementary video.

Table 4: **Effects of image transfer modules.**        Table 5: **Effects of fusion losses.**

| Methods | RMSE↓ | Hausdorff↓ | STED ↓ | Methods | RMSE↓ | Hausdorff↓ | STED ↓ |
|---|---|---|---|---|---|---|---|
| w/o Body Attn. | 7.25 | 26.54 | 0.0356 | Init. (no normal) | 9.98 | 30.04 | 0.0771 |
| ResNet Encoder | 6.55 | 25.01 | 0.0331 | Init. + $\mathcal{L}_e$ | 5.52 | 23.45 | 0.0525 |
| w/ Cross Modal | 5.13 | 22.78 | 0.0264 | Init. + $\mathcal{L}_e$ + $\mathcal{L}_r$ | 4.96 | 22.23 | 0.0238 |
| **Full Model** | **4.66** | **20.89** | **0.0205** | **Full Model** | **4.66** | **20.89** | **0.0205** |

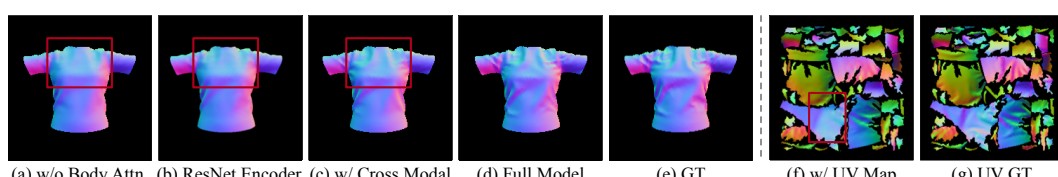

(a) w/o Body Attn.   (b) ResNet Encoder   (c) w/ Cross Modal   (d) Full Model   (e) GT   (f) w/ UV Map   (g) UV GT

Figure 6: **Comparison of design variants for image transfer modules.** We show that design variants in the network architecture (a), (b), (c) produce smoother results, while the full model (d) can generate fine wrinkles that are closer to the GT (e). In addition, we show that using automatically generated UV maps (f) results in complex islands that are not beneficial for wrinkle estimation.

**Effects of Fusion Losses.** In Table 5 and Figure 7, we show the effect of each loss during fusion optimization. The initial mesh (a) interpolated from the position images does not contain enough high-frequency wrinkle details, and non-visible vertices at side views can not be constrained, which leads to large RMSE error. By enforcing edge length consistency, we repair non-visible vertices as in (b). Jointly with (a) and (b), we obtain a reasonably good initialization that allows normal fusion to be feasibly achieved with few optimization steps. Moreover, we recover more accurate wrinkles after

fusing from normal images (c). However, with only the normal loss, we observe artifacts at mesh rims due to under-constrained objectives. We thus further refine the results by penalizing irregular boundary edges and abrupt normal changes, which generate smoothed results as in (d). In summary, by including all losses during fusion, we can generate accurately deformed garments with higher perceptual quality. We include more ablation studies on the collision loss in Appendix E.

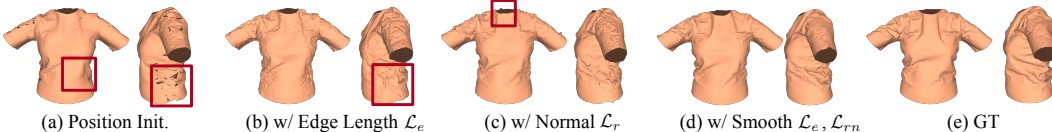

(a) Position Init.    (b) w/ Edge Length $\mathcal{L}_e$    (c) w/ Normal $\mathcal{L}_r$    (d) w/ Smooth $\mathcal{L}_e, \mathcal{L}_{rn}$    (e) GT

Figure 7: **Illustration of intermediate fusion results.** From left to right, we show optimization results after adding corresponding losses. By fusing both modalities, we produce more accurate deformation with higher perceptual quality. Moreover, the edge length and normal consistency help to constrain non-visible vertices and resolve artifacts at mesh rims.

### 4.6 GENERALIZATION EVALUATION

In all above evaluations, we evaluate on a single garment instance each time to ensure a fair comparison with baseline methods (Santesteban et al., 2019; 2021; Pan et al., 2022; Zhao et al., 2023). However, our method is not limited to a single garment input and is well-suited for training across multiple garments thanks to several designs: (*i*) the use of pretrained DINO encoder that is capable of extracting detailed semantic features for various garments, (*ii*) the image transfer approach that is agnostic to garment topologies, and (*iii*) the use of front and back view projections to establish image representations for garments, which will not be scalable for manual UV parameterization on a large collection of garments. To verify the generalizability of our method, we further jointly train on 50 dress garments on the CLOTH3D (Bertiche et al., 2020a) dataset, and show the results in Figure 8. By training across multiple garments, our method can effectively generalize to unseen garment shapes, with the skinning-free method particularly beneficial for tackling loose garment deformation.

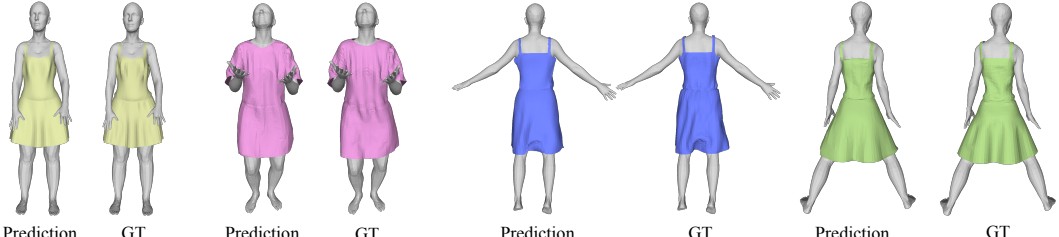

Prediction       GT          Prediction       GT          Prediction       GT          Prediction       GT

Figure 8: **Results on unseen garments.** We show more results on unseen loose garments in the CLOTH3D (Bertiche et al., 2020a) to verify the generalizability and scalability of our method.

## 5 DISCUSSION

**Limitation.** Although our method succeeds in generating accurate deformation on common garments, it assumes the garment template has a single layer and is flat. For multi-layered garments, the rendered images may not capture inner garments due to occlusion in perspective projection. We also only model body-garment interaction in cross-attention and do not consider interaction among garment layers. Furthermore, we tackle a single input body to reduce model complexity, while future works are encouraged to explore encoding for body motions in the image transfer network. More discussions on failure cases and societal impact are included in Appendix G.

**Conclusion.** In this paper, we propose a novel skinning-free pipeline to generate accurate 3D garment deformation via image transfer. We decompose garment deformation into decoupled frequency modalities represented by vertex positions and normals, and further project both modalities into the image space, which allows us to leverage pretrained image encoders and body-garment cross-attention to recover pose-conditioned fine wrinkles of higher perceptual quality. Thanks to these designs, our method effectively produces more accurate wrinkle details over previously dominant skinning-based baselines, and the proposed pipeline can be broadly applied to generate detailed geometry deformation with mixed frequency modalities on other manifolds.

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

## A    DATASET

The licenses for VTO (Santesteban et al., 2019), TailorNet (Patel et al., 2020) and CLOTH3D (Bertiche et al., 2020a) datasets can be found in the below urls:

**VTO**: `https://github.com/isantesteban/vto-dataset`.

**TailorNet**: `https://github.com/zycliao/TailorNet_dataset`.

**CLOTH3D**: `https://chalearnlap.cvc.uab.cat/dataset/38/description/`.

## B    IMAGE RENDERING

We illustrate the rendering process as shown in Figure 9. For a deformed mesh, we convert its vertex positions or normals to RGB *colors*, while projecting the garment template mesh with its *vertices* to render corresponding images in both front and back views. Note that the image silhouette remains the same for all deformed meshes as we always project the template vertices. From the rendered images, we observe that the position images mostly contain low-frequency information, *e.g.* areas of colors representing the posed garment shape, while the normal images mostly contain high-frequency details such as edges for wrinkles. Motivated this observation, we propose to leverage both modalities to model garment deformation, which facilitates to generate accurately deformed garments.

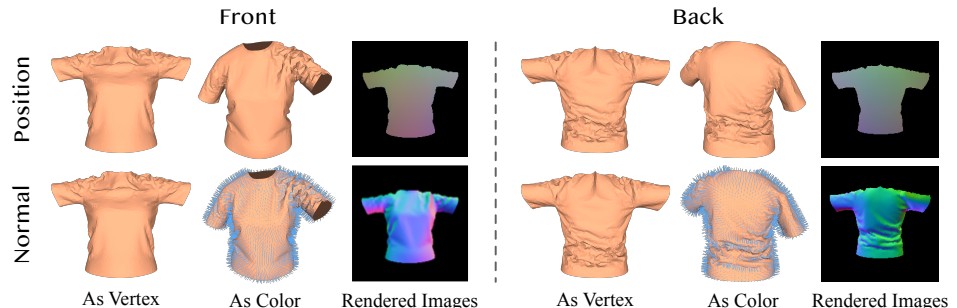

Figure 9: **Illustration of image rendering.** We project template mesh with its *vertices* and convert geometric attributes into *colors* to render images from both front and back views.

## C    NETWORK ARCHITECTURE

We use the DINO (Caron et al., 2021) encoder model `dino-vitb16` pretrained on the ImageNet (Deng et al., 2009). During training, we fine-tune the last two layers `10` and `11`, as well as the final `layernorm` module. The encoded image features $\bar{\mathbf{F}}_g^s, \mathbf{F}_b^s$ are in the shape $\mathbb{R}^{257 \times 768}$, where we include the `[CLS]` token to distinguish garment and body inputs. In each transformer block in the pose-conditioned feature refinement module, we refine the input garment feature $\mathbf{F}_g^{(0)}$, ignoring the superscript of view $s$ for simplicity, as:

$$\mathbf{F}_g^{(1)} = \mathbf{F}_g^{(0)} + \text{Cross}(\text{Norm}(\mathbf{F}_g^{(0)}), \mathbf{F}_b^s, \mathbf{F}_b^s) \tag{9}$$

$$\mathbf{F}_g^{(2)} = \mathbf{F}_g^{(1)} + \text{Self}(\text{Norm}(\mathbf{F}_g^{(1)}), \text{Norm}(\mathbf{F}_g^{(1)}), \text{Norm}(\mathbf{F}_g^{(1)})) \tag{10}$$

$$\mathbf{F}_g^{(3)} = \mathbf{F}_g^{(2)} + \text{MLP}(\text{Norm}(\mathbf{F}_g^{(2)})) \,, \tag{11}$$

where $\mathbf{F}_g^{(1)}, \mathbf{F}_g^{(2)}$, and $\mathbf{F}_g^{(3)}$ are intermediate output features, $\text{Cross}(\cdot)$ and $\text{Self}(\cdot)$ denote 4-heads of cross and self attention respectively, and $\text{Norm}(\cdot)$ denotes the layer norm. In the MLP, we use linear layers of [3072, 768] and GELU activation. Finally, we use $L = 4$ residual blocks of 2D convolutions to construct the image decoder. In each block, we decode the input feature $\hat{\mathbf{F}}_g^{(0)}$ as:

$$\hat{\mathbf{F}}_g^{(1)} = \text{Conv}_1(\hat{\mathbf{F}}_g^{(0)}) + \text{Conv}_2(\text{NL}(\text{Conv}_3(\text{NL}(\hat{\mathbf{F}}_g^{(0)})))) \tag{12}$$

$$\hat{\mathbf{F}}_g^{(2)} = \text{TransConv}(\hat{\mathbf{F}}_g^{(1)}) \,, \tag{13}$$

where $\text{Conv}_i(\cdot)$ denotes the 2D convolution layer. For $i = \{1, 3\}$, the convolution halves the feature dimension, while for $i = 2$, the output dimension remains the same as the input. $\text{NL}(\cdot)$ denotes the non-linear Swish activation function, and $\text{TransConv}(\cdot)$ denotes the transposed 2D convolution that doubles the spatial resolution of the features.

## D  INFERENCE TIME COMPARISON

We compare inference time on the "T-shirt" and "Dress" garments in the VTO dataset, which contain 4K and 12K triangles respectively. While we use test-time optimization for multimodal fusion, our method is significantly faster than simulation-based method (Narain et al., 2014) and comparable with physics-based methods (Grigorev et al., 2023), thus maintaining its practical applicability. Since we directly obtain the posed garment shape from pixel values of transferred position images as initialization, which is close to the optimal results and helps to improve convergence speed. In addition, the normal optimization is relatively simple and well-conditioned. These two designs ensure the efficiency of the fusion process and allows the optimization to converge in only 100 steps.

Table 6: **Inference time comparison.** Our method is more efficient than simulation (Narain et al., 2014) and physics (Grigorev et al., 2023) based methods.

| Time (s) | (Narain et al., 2014) | (Grigorev et al., 2023) | **Ours** | (Patel et al., 2020) | (Santesteban et al., 2021) | (Santesteban et al., 2019) |
|---|---|---|---|---|---|---|
| T-shirt | 3.891 | 0.127 | 0.115 | 0.028 | 0.003 | 0.005 |
| Dress | 5.680 | 0.167 | 0.153 | 0.040 | 0.004 | 0.008 |

## E  MORE ABLATION STUDIES

**Alternative Networks and Input Designs.** We compare our method with baselines that directly estimate geometric attributes using 3D networks: PointNet (Qi et al., 2017) and GCN (Xu et al., 2018). Due to limited network capacity, both networks fail to achieve superior accuracy compared to our image-based approach. This justifies the efficacy of the proposed method to model garment deformation via image transfer. In addition, we compare with variants that use automatically generated UV maps as image inputs. As shown in Figure 6, such UV images contain a large number of islands that destroy the garment shape priors, which do not provide meaningful semantic information for the image feature extraction. In contrast, the proposed method achieves the best accuracy over all design variants, as quantitatively compared in Table 7.

Table 7: **Variants of networks and inputs.**

| Methods | RMSE↓ | Hausdorff↓ | STED ↓ |
|---|---|---|---|
| PointNet | 9.78 | 27.76 | 0.0495 |
| GCN | 10.13 | 28.05 | 0.0503 |
| UV Maps | 9.01 | 26.52 | 0.0425 |
| **Ours** | **4.66** | **20.89** | **0.0205** |

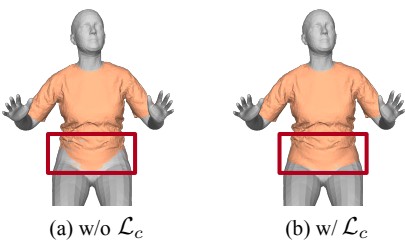

(a) w/o $\mathcal{L}_c$      (b) w/ $\mathcal{L}_c$

Figure 10: **Effects of collision loss.**

**Effects of Collision Loss.** As shown in Figure 10, for challenging body poses, failure in regressing accurate positions can cause garments to inter-penetrate the body. To avoid such artifacts, we include $\mathcal{L}_c$ to resolve body-garment collision, which effectively generates penetration-free results.

## F  MORE QUALITATIVE RESULTS

In this section, we show more results with challenging poses and body shapes in Figure 11. Since high frequency wrinkle details are effectively decoupled, our method can consistently produce high-fidelity results on challenging poses. In the main paper, we compare on the $\beta = 0$ body shape to align with the train-test split from baseline methods. For completeness, we show that our method can

generalize to unseen body shapes when jointly trained with multiple body shapes. Moreover, we compare in Figure 12 that under this setting, our method can adapt to different body shapes to produce correspondingly plausble results.

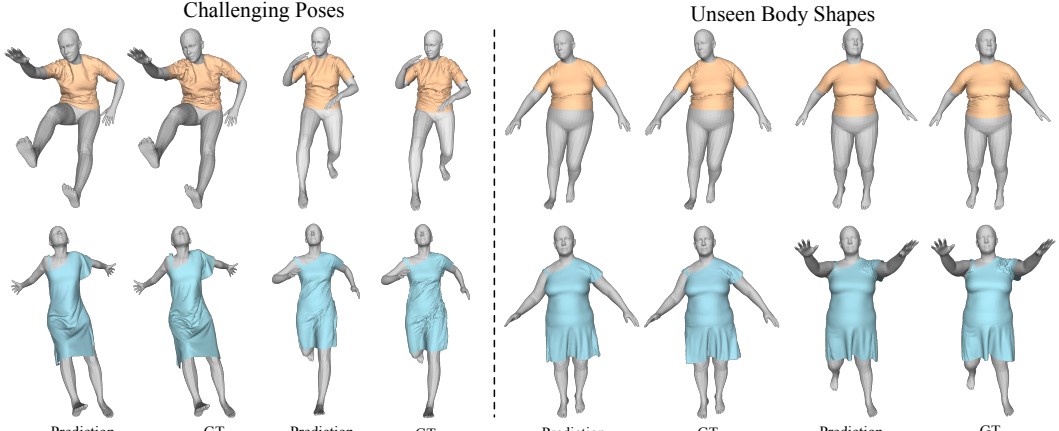

Figure 11: **Generalization to challenging poses and unseen body shapes.** Our method can consistently produce accurate and detailed wrinkles on challenging poses and unseen body shapes.

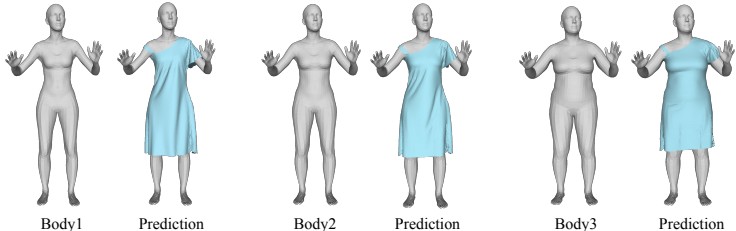

Figure 12: **Results on different body shapes.** Our method can produce different garment deformations conditioned on different body shapes.

## G LIMITATION AND SOCIETAL IMPACT

**Failure Case.** To trade for model efficiency, we only use front and back views to render the images, with the assumption that common garment templates are flat under these two views and are reasonably thin. For non-visible side views, the vertex positions are constrained by the edge length and normal consistency losses $\mathcal{L}'_e$ and $\mathcal{L}_{rn}$, respectively. However, due to the lack of direct supervision on these vertices, their deformation may not align with the ground truth data and contain undesired artifacts, as shown in Figure 13. We encourage future works to explore more robust approaches to estimate side view deformations, in particular leverage learning-based methods.

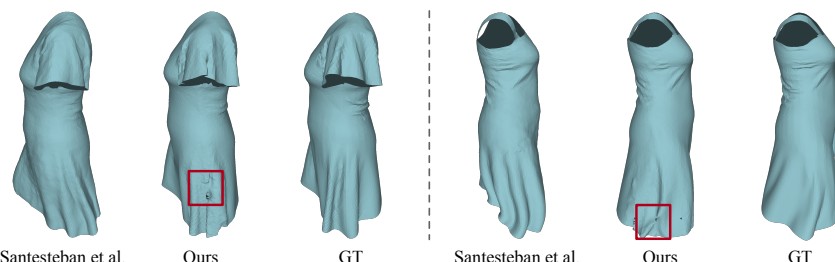

Figure 13: **Illustration of Failure Cases.** Failure to constrain non-visible side view vertices can produce incorrect deformations with undesired artifacts.

**Societal Impact.** Since we rely on a learning-based method to produce garment deformations, when applied to unseen poses, it may fail to produce correct garment deformation and thus dressing the character in an inappropriate manner, which are not suitable to display for public.

