# OpenReview forum: "Skinning-free Accurate 3D Garment Deformation via Image Transfer"
_ICLR.cc/2025/Conference — ICLR 2025 Conference Withdrawn Submission_

### Official Review · Reviewer_JcCp · 2024-11-02

**Soundness:** 3
**Presentation:** 3
**Contribution:** 3
**Rating:** 6
**Confidence:** 3

**Summary:**

This paper presents a method that involves a vertex position and normal prediction and an optimization that generate deformed garments according to the posed body shape. They demonstrated better deformed garments compared to previous method.

**Strengths:**

* The 2D image encoding seems working pretty well on the vertex position and normal prediction.
* The optimization formulation is reasonable and achieve good result.
* The ablation study demonstrate the effectiveness of each components, give us some insight on the design decision.

**Weaknesses:**

* Although some limitations are discussed, it is better to list and show broken and failing cases in the paper specifically. For example, maybe show results for extreme poses or unusual garment types.
* The comparison method seems not cover some latest works, e.g., [Zhang et al. 2022]: Motion Guided Deep Dynamic 3D Garments, SIGGRAPH ASIA 2022.

**Questions:**

* Although it sounds reaosnable, I am still wondering what will be the result if the proposed method use a different 2D representation? e.g., directly use the 2D projection of the 3D shape instead of encoding position and normal into RGB?
* In the current version, there are only two views are used. I am wondering will it be helpful to add more views, e.g., from two different side views? And why the authors choose not to use them in the first place? Specifically, I recommend the authors to discuss the trade-offs considered when choosing the number of views, such as computational cost versus accuracy, and whether any experiments were conducted with different numbers of views.

---

### Official Review · Reviewer_SSYd · 2024-11-02

**Soundness:** 2
**Presentation:** 3
**Contribution:** 2
**Rating:** 3
**Confidence:** 4

**Summary:**

The paper proposes a framework for neural 3D garment deformation, which decomposes the low- and high-frequency modalities of a deformation field into a displacement field and a normal field. The novelty lies in its predictive learning representation: instead of directly predicting 3D deformations, the method predicts front and back views of displacement and normal renderings, conditioned on the displacement and normal renderings of the posed human body. The deformed 3D garment is then reconstructed by deforming its initial template based on the predicted displacement and normal maps under the edge-length elasticity and normal consistency constraints.

**Strengths:**

- The idea of representing deformation using 2D images is novel and inspiring because 2D images are more structured than 3D data and may be more easy to learn.
- The 3D fusion algorithm, i.e., deforming 3D mesh based on displacement renderings and normal renderings, may have wider applications.

**Weaknesses:**

The method uses statics to approximate dynamics, assuming that deformation is entirely determined by human pose. While this assumption holds for draping, it does not apply to humans in motion, as it completely omits the inertia effects of cloth. The garment's current state should depend on its previous state. Omitting inertia can cause **jittering artifacts**  because the stable equilibria of garments are sensitive to collision boundary conditions. This is evidenced in the supplementary video, where garment vibrations are much more visible compared to previous methods.

**Questions:**

The authors claim "collision-free" results in Figure 5; however, there are actually more penetrations between the cloth and body compared to previous methods in the supplementary video.

---

### Official Review · Reviewer_xSQM · 2024-11-03

**Soundness:** 3
**Presentation:** 3
**Contribution:** 3
**Rating:** 5
**Confidence:** 4

**Summary:**

This paper proposes  a deep learning based approaches to draping virtual 3D garments over bodies in various poses. The problem is framed as an image to image translation task: garments are represented in image space, from the front at the back.

**Strengths:**

- The paper is technically sound and very clear
- Garment deformations are represented at 2 levels of detail
    - Coarse, by predicting vertex positions
    - Finer, with a normal map
- An optimization fuses information from both views and the normal maps in a consistent manner.
- Results appear to be quantitatively and qualitatively superior to existing methods.

**Weaknesses:**

The main issue relates to the whole motivation of tackling garment draping with a neural network based approach. Existing papers build and demonstrate on at least one of the following aspects:
- computational efficiency, with network based approaches being orders of magnitude faster and less computationally demanding than classical simulators.
- Differentiability, to solve inverse problems - such as recovering garment geometries from real images.
The present paper does not present any benefit related to this. Its pipeline is not lightweight, nor differentiable.
The introduction mentions that traditional garment simulators are time consuming and hard to set up. However, the present method relies on simulation data to be trained, and does not show any sign of generalization beyond its training data. In other words, it does not alleviate the issues of traditional simulators, since it builds upon them.

Another limitation is the limited generalization: single layered garments, with simple topology, over a single body shape.

**Questions:**

Can the proposed approach demonstrate its benefits regarding either:
- computation efficiency
- Or differientiability
over existing and classical approaches?

Other question:
- how is temporal consistency handled over successive timeframes?

---

### Official Review · Reviewer_YvvF · 2024-11-07

**Soundness:** 2
**Presentation:** 2
**Contribution:** 2
**Rating:** 3
**Confidence:** 5

**Summary:**

This paper presents a skinning-free approach to generate accurate 3D garment deformation via image transfer. This paper formulates garment deformation into decoupled high-low frequency garment vertex position and low-frequency vertex
normal for high-frequency local wrinkle details and represent them in the images space to leverage pre-trained image encoders. However, the authors missed discussing one important paper and comparisons. Some of the key details also missing in the paper. Please see the weakness section.

**Strengths:**

Please see the weakness section.

**Weaknesses:**

-	It is difficult to encode garment geometry of significantly different size of garments within the same resolution of image e.g. a long ankle length skirt (10K vertices) vs a short top(200 Vertices). This indicates the limitations of the method to simple small size of the garments. It would be better if authors can show results on a wider range of garment sizes or discuss how their method might be extended to handle very large garments.
-	Another issue is, how the self-occlusion of garments is going to be captured in the images. i.e How does your method handle self-occlusion of garments, particularly for encoding the geometry of occluded vertices under perspective projections?
-	To alleviate the above two problems, one earlier method DeepDraper ICCV 2021, followed both image and explicit geometry (offset) based approach. Predicting normals and learning via losses on normal mapped texture images has been proposed in the DeepDraper paper, where they rendered front, back, sides viewpoints. DeepDraper showed improved results over TailorNet, also elevating the requirements of training individual high frequency and low frequency modules like TailorNet.
-	The authors did not discuss DeepDraper paper, which I recommend authors to look at and add comparison results.
-	Considered only a single body, is very hard constraint. It is difficult to say how sensitive is method for varying body sizes whether it will work or not. Can the authors demonstrate that their method can handle wide variety of the body meshes in a single trained model ?
-	Could you clarify whether your models are trained separately for each garment type, or if a single model is trained for all garment types?
-	The collision loss used is similar to what generally used, yet there is no collision seen in any of the results which seems strange and raise question on selection of examples in the videos. If authors have done any post-processing for collision handling, then the results before and after post processing should have been shown in the results. Can the authors please elaborate their collision handling strategy in detail, whether they are doing any post processing?.
- The video comparison is very limited, a comparison with HOOD which is the closest best as we can see in the tables must be added. Also, since the paper technically resembles to DeepDraper in terms for using normal textures images, a comparison with the same also must. I recommend authors to show a video comparison with HOOD and DeepDraper, with varying garment sizes, types and body shapes and sizes.

**Questions:**

Please see the weakness section.

---

### Note · Authors · 2024-11-15

I have read and agree with the venue's withdrawal policy on behalf of myself and my co-authors.